# Quercetin Enhances the Thioredoxin Production of Nasal Epithelial Cells In Vitro and In Vivo

**DOI:** 10.3390/medicines5040124

**Published:** 2018-11-21

**Authors:** Yukako Edo, Amane Otaki, Kazuhito Asano

**Affiliations:** 1Graduate School of Health Sciences, Showa University Graduate School, Yokohama 226-8555, Japan; yukakoeddy18@gmail.com; 2Division of Nursing, Showa University School of Nursing and Rehabilitation Sciences, Yokohama 226-8555, Japan; aotaki@nr.showa-u.ac.jp; 3Division of Physiology, Showa University School of Nursing and Rehabilitation Sciences, Yokohama 226-8555, Japan

**Keywords:** allergic rhinitis, mice, quercetin, thioredoxin, nasal epithelial cell, production, increase, in vitro, in vivo

## Abstract

**Background**: Thioredoxin (TRX) acts as both a scavenger of reactive oxygen species (ROS) and an immuno-modulator. Although quercetin has been shown to favorably modify allergic rhinitis (AR) symptoms, its influence on TRX production is not well defined. The present study was designed to examine whether quercetin could favorably modify AR symptoms via the TRX production of nasal epithelial cells in vitro and in vivo. **Methods**: Human nasal epithelial cells (HNEpCs) were stimulated with H2O2 in the presence of quercetin. TRX levels in 24-h culture supernatants were examined with ELISA. BALB/c male mice were intraperitoneally sensitized to ovalbumin (OVA) and intranasally challenged with OVA every other day, beginning seven days after the final sensitization. The mice were orally administered quercetin once a day for five consecutive days, beginning seven days after the final sensitization. Nasal symptoms were assessed by counting the number of sneezes and nasal rubbing behaviors during a 10-min period immediately after the challenge. TRX levels in nasal lavage fluids obtained 6 h after the challenge were examined by ELISA. **Results**: Treatment with 1.0 nM quercetin increased H2O2-induced TRX levels. The oral administration of 20.0 mg/kg of quercetin significantly inhibited nasal symptoms after the challenge. The same dose of quercetin significantly increased TRX levels in nasal lavage fluids. **Conclusions**: Quercetin’s ability to increase TRX production may account, at least in part, for its clinical efficacy toward AR.

## 1. Introduction

Allergic rhinitis (AR) is a well-known type of chronic allergic inflammation that occurs in the nasal mucosa and is characterized by multiple symptoms such as sneezing, itching, and watery rhinorrhea [1,2]. Although AR is not life-threatening, it places a significant burden on patients and society because its symptoms lead to inconveniences in daily life. These clinical symptoms also exert adverse effects on industrial work productivity and school learning performance, resulting in increased medical costs and lower quality of life [1,2].

AR treatment can be divided into three main categories: allergen avoidance, drug therapy, and immunomodulating therapy [3]. Allergen avoidance is the safest mode of treatment, but it is often insufficient to obtain satisfactory results [3]. Although histamine H1 receptor antagonists and topical steroids can significantly ease the associated symptoms, they require repeated treatment sessions over the patient’s lifetime [3,4]. Moreover, the currently available therapeutic agents cause adverse effects, including dizziness, dry mouth, and constipation [3,4]. Although immunotherapy induces immunological tolerance through the subcutaneous injection of or sublingual application of allergens, it has several disadvantages: it requires several years of therapy, is expensive, and contains a risk of anaphylaxis [5]. Furthermore, many patients dislike taking daily medication merely for prevention [3]. Therefore, the development of new medications for the treatment of allergic diseases, including AR, is desired.

Quercetin is a dietary flavonoid found in red wine, tea, many fruits, and onions [6]. For many years, the possible healthy biological activities of quercetin have been studied, with anti-pollinosis, anti-diabetic, and anti-viral activity reported [7]. Moreover, quercetin acts as a scavenger of free radicals, which damage cell membranes, tamper with DNA, and even cause cell death [8,9,10]. Quercetin also plays a role in allergic inflammatory responses by inhibiting mast cells and eosinophils from producing chemical mediators (e.g., histamine and leukotriene) and inflammatory cytokines, which are responsible for the induction and persistence of allergic reactions [11,12]. Furthermore, the oral administration of quercetin can alleviate ocular and nasal symptoms observed in patients with pollinosis [13]. Quercetin’s attenuating effect on the clinical symptoms of allergic reactions has also been observed in experimental animal models of allergic asthma and AR [14,15,16]. Although these reports strongly suggest that quercetin is a good dietary supplement candidate for preventing the development of allergic diseases such as AR, the precise mechanisms by which quercetin modulates the clinical symptoms of allergic diseases remain unknown.

It is currently accepted that inflammatory cells including eosinophils, which are the most important effector cells in the development of inflammatory diseases, produce several types of toxic granule proteins and reactive oxygen species (ROS), such as O_2_ and H_2_O_2_ [17,18]. Although the physiological production of ROS is generally considered essential in host defense and to maintain homeostasis, the overproduction of ROS and their metabolites are harmful and cause oxidative stress responses, which are implicated in the pathogenesis of allergic inflammatory airway diseases, including AR [17,19]. Conversely, under normal physiological conditions, several types of endogenous antioxidants, such as glutathione and superoxide dismutase, prevent the development of oxidative stress responses [19]. Among these, thioredoxin (TRX) has attracted attention as an endogenous antioxidant protein. TRX is a 12-kDa protein with two redox (reduction/oxidation) active half-cysteine residues [20,21]. In addition to its anti-oxidative activity, TRX is reported to exert immunomodulatory effects. The administration of exogenous TRX suppresses airway hyperresponsiveness induced by specific allergens by inhibiting eosinophil accumulation in the airways of asthmatic mouse models [22,23]. TRX has also been reported to augment the production of Th1-type cytokines, such as IL-12 and IFN-γ, which prevent allergic responses. These reports suggest that manipulating TRX production may be a good target for the treatment of chronic airway allergic diseases, including AR [22,23]. However, the influence of quercetin on the production of TRX is currently unclear. Therefore, the present study investigated the influence of quercetin on the TRX system by examining the ability of agents from human nasal epithelial cells (HNEpCs) to affect TRX production in vitro and in vivo.

## 2. Materials and Methods

### 2.1. Mice

Specific 5-week-old pathogen-free BALB/c male mice were purchased from CLEA Japan Co., Ltd. (Tokyo, Japan). The mice were maintained in our animal facility at 25 °C ± 2 °C with 55% ± 10% humidity under a 12-h dark/light cycle and were allowed free access to tap water and standard laboratory rodent chow (Oriental Yeast Co., Ltd., Tokyo, Japan) throughout the experiments. Each control and experimental group consisted of five mice. All animal experiments were approved by the Ethics Committee for Animal Experiments of Showa University (Approved No. 54011). Date of approval: 1 April 2018.

### 2.2. Reagents

Quercetin was purchased from Sigma-Aldrich Co., Ltd. (St. Louis, MO, USA) as a preservative-free pure powder. Quercetin was first dissolved in dimethyl sulfoxide (DMSO) at a concentration of 10.0 mM. This solution was then diluted with Airway Epithelial Cell Growth Media (AECG medium; PromoCell GmbH, Heidelberg, Germany) at appropriate concentrations for the experiments; then, the solution was sterilized by passing through 0.2-μm filters and was stored at 4 °C until use. To assess in vivo use, quercetin was mixed with 5% tragacanth gum solution at a concentration of 7.5 mg/mL [13]. Chicken ovalbumin (OVA; grade V) and Al(OH)_3_ (alum) were obtained from Sigma-Aldrich Co. Ltd. as preservative-free pure powders.

### 2.3. Cell Culture

HNEpCs, purchased from PromoCell GmbH, were suspended in AECG medium (PromoCell GmbH) at a concentration of 5 × 10^5^ cells/mL and used as target cells. The HNEpCs (5 × 10^5^ cells/mL) were stimulated with 12.5–100.0 μM H_2_O_2_ for 24 h in a final volume of 2.0 mL. To examine the influence of quercetin on TRX production, the HNEpCs (5 × 10^5^ cells/mL) were stimulated with 50.0 μM H_2_O_2_ in the presence of 0.1–10.0 nM quercetin for 24 h in a final volume of 2.0 mL. To examine TRX mRNA expression, the cells were stimulated with 50.0 μM H_2_O_2_ in the presence of 0.1–10.0 nM quercetin for 12 h. Quercetin was added to the cell cultures 2 h before H_2_O_2_ stimulation.

### 2.4. Assay to Assess Cytotoxicity of H_2_O_2_ and Quercetin

HNEpCs (5 × 10^5^ cells/mL) were cultured with either 12.5–100.0 μM H_2_O_2_ or 0.1–10.0 nM quercetin for 24 h. The cells were then collected, and cell viability was assessed by the trypan blue dye exclusion test. The dead cells were stained with trypan blue, and the proportion of dead cells was determined by counting 300 total cells.

### 2.5. Assay to Assess TRX mRNA Expression

TRX mRNA expression was examined by the methods described previously (24). Briefly, Poly A^+^ mRNA was extracted from cultured cells with oligo(dT)-coated magnetic micro beads (Milteny Biotec, Bergisch Gladbach, Germany). The first-strand cDNA was synthesized from 1.0 μg of Poly A^+^ mRNA using a Superscript cDNA synthesis kit (Invitrogen Corp., Carlsbad, CA, USA) according to the manufacturer’s recommendations. Polymerase chain reaction (PCR) was then performed using a GeneAmp 5700 Sequence Detection System (Applied Biosystems, Forster City, CA, USA). The PCR mixture consisted of 2.0 µL of sample cDNA solution (100 ng/µL), 25.0 µL of SYBR-Green Mastermix (Applied Biosystems), 0.3 µL of both sense and antisense primers, and distilled water for a final volume of 50.0 µL. The conditions used for the reaction was as follows: 4 min at 94 °C, followed by 40 cycles of 15 s at 95 °C and 60 s at 60 °C. GAPDH was used as an internal control. TRX mRNA levels were calculated using the comparative parameter threshold cycle and normalized to GAPDH. The primers used for real-time RT-PCR were as follows: 5′-GCCTTGCAAAATGATTCAAGC-3′ (Sense) and 5′-TTGGCTCCAGAAAATTCACC-3′ (Antisense) for TRX [24], and 5′-TGTTGCCATCAATGACCCCTT-3′ (Sense) and 5′-CTCCACGACGTACTCAGCG-3′ (Antisense) for GAPDH [24].

### 2.6. Sensitization and Treatment of Mice

BALB/c mice were sensitized with an intraperitoneal injection of 20.0 μg/mL OVA in phosphate-buffered saline (PBS) combined with 1.0 mg of alum in a total volume of 200.0 μL on days 0, 7, and 14 [3,4]. On days 21, 23, and 25, the mice were intranasally instilled with 100 μg of OVA (5.0 μL in PBS) [3,4]. The mice were orally administered 10, 15, 20, or 25 mg/kg of quercetin using a stomach tube in a volume not exceeding 0.5 mL once a day for five consecutive days, beginning on day 21 relative to the sensitization.

### 2.7. Collection of Nasal Lavage Fluids

The mice were anesthetized by intraperitoneal injection with 50.0 mg/kg sodium pentobarbital (Kyoritsu Seiyaku Co., Ltd., Tokyo, Japan) 6 h after the OVA nasal challenge. The trachea was exposed and cannulated to introduce 1.0 mL of PBS [16]. The lavage fluid from the nares was collected and centrifuged at 3000 rpm for 15 min at 4 °C. After measuring IgA levels with ELISA (Bethyl Lab., Inc., Montgomery, TX, USA), the fluids were stored at −40 °C until use [16].

### 2.8. Assessment of Nasal Symptoms

Nasal allergy symptoms were assessed by counting the number of sneezes and nasal rubbing movements for 10 min immediately after the OVA nasal instillation. The experimental mice were placed into plastic animal cages (35 × 20 × 30 cm) for approximately 10 min to acclimate. After the nasal instillation of 0.1% OVA solution in PBS in a volume of 5.0 μL, the mice were placed into plastic cages (two animals/cage), and the number of sneezes and nasal rubbing movements were counted for 10 min [16].

### 2.9. TRX Assay

The TRX levels in the culture supernatants and nasal lavage fluids were examined using human and mouse TRX ELISA test kits (CUSABIO TECHNOLOGY LLC., Huston, TX, USA) according to the manufacturer’s recommendations. The minimum detectable levels of the ELISA test kits were 1.172 ng/mL and 0.078 ng/mL for humans and mice, respectively.

### 2.10. Oxidative Stress Assay

The oxidative stress responses in the nasal mucosa were evaluated by measuring lipid peroxide levels in nasal lavage fluids using d-ROM tests (DIACRON, Via Zircone, Italy) according to the manufacturer’s recommendations. The results were expressed as mean Carratelli Units (CARR U) ± SE.

### 2.11. Statistical Analysis

The statistical significance between the control and experimental groups was assessed with an ANOVA followed by Dunette’s multiple comparison test. A *P* value of less than 0.05 was considered significant.

## 3. Results

### 3.1. Influence of H_2_O_2_ Stimulation on TRX Production from HNEpCs in Vitro

The first experiments were performed to examine whether H_2_O_2_ stimulation could increase TRX production from HNEpCs and to determine the optimal concentration of H_2_O_2_ for stimulation. Thus, the cells were stimulated with various concentrations of H_2_O_2_ for 24 h, and the TRX levels in the culture supernatants were determined via ELISA. As shown in Figure 1, the stimulation of cells with H_2_O_2_ caused a significant increase in the ability of cells to produce TRX. As little as 2.5 μM H_2_O_2_ caused a strong stimulation in TRX production. Maximum production was observed with 25.0–75.0 μM H_2_O_2_ whereas 100.0 μM H_2_O_2_ was inhibitory (Figure 1).

### 3.2. In Vitro Influence of Quercetin on H_2_O_2_-Induced TRX Production from HNEpCs

The second set of experiments was designed to examine the influence of quercetin on the TRX production of HNEpCs after H_2_O_2_ stimulation. The cells were stimulated with 50.0 μM H_2_O_2_ in the presence or absence of quercetin for 24 h. TRX levels in the culture supernatants were examined by ELISA. As shown in Figure 2, the treatment of cells with quercetin at concentrations of both 0.1 nM and 0.5 nM barely affected the ability of the cells to produce TRX: the TRX levels in the culture supernatants were nearly identical (not significant) to those detected in the controls. At concentrations greater than 1.0 nM, however, quercetin induced significantly increased TRX levels in culture supernatants compared to those levels in the controls.

### 3.3. Influence of H_2_O_2_ and Quercetin on Cell Viability

The third set of experiments was performed to examine the influence of H_2_O_2_ and quercetin on cell viability. HNEpCs were cultured with either H_2_O_2_ or quercetin for 24 h, and cell viability was examined via the trypan blue dye exclusion test. Although the cells cultured with H_2_O_2_ concentrations less than 50.0 μM did not display reduced cell viability, 100.0 nM H_2_O_2_ caused significant cell death (Figure 3A). We then examined the influence of quercetin on cell viability. Quercetin did not exert cytotoxic effects on HNEpCs; the number of dead cells observed in cells cultured with 100.0 nM quercetin was nearly identical to that observed in controls (Figure 3B).

### 3.4. Influence of Quercetin on TRX mRNA Expression

The fourth set of experiments was performed to examine the influence of quercetin on TRX mRNA expression in HNEpCs stimulated with 50.0 μM H_2_O_2_. The stimulation of HNEpCs with H_2_O_2_ caused significant increases in TRX mRNA expression compared to the non-stimulated (Med. alone) controls (Figure 4). However, TRX mRNA expression was significantly suppressed in HNEpCs treated with more than 1.0 nM quercetin but not in HNEpCs treated with less than 0.5 nM, whereas TRX mRNA expression was increased by stimulation with H_2_O_2_ (Figure 4).

### 3.5. Influence of Quercetin on Oxidative Stress Responses in Nasal Mucosa

The fifth set of experiments was performed to examine whether oxidative stress responses were occurred in OVA-sensitized mice and whether quercetin administration into OVA-sensitized mice could modulate oxidative stress responses. Therefore, OVA-sensitized mice were orally administered 10.0–25.0 mg/kg of quercetin at days 21–25 after sensitization. Nasal lavage fluids were obtained 6 h after the final nasal OVA challenge, and lipid peroxide levels in nasal secretions were examined by the d-ROM test. Quercetin treatment significantly decreased lipid peroxide levels in the nasal lavage fluids of the mice, whereas the OVA nasal challenge increased lipid peroxide levels (Figure 5).

### 3.6. Influence of Quercetin on the Appearance of TRX in Nasal Lavage Fluids

The sixth set of experiments was designed to examine the influence of quercetin on the appearance of TRX in nasal lavage fluids obtained from sensitized mice after the OVA nasal challenge. The OVA-sensitized mice were orally administered 10.0–25.0 mg/kg of quercetin at days 21–25 after sensitization. Nasal lavage fluids were obtained 6 h after the final nasal OVA challenge. As shown in Figure 6, the oral administration of 20.0 and 25.0 mg/kg of quercetin, but not 10.0 and 15.0 mg/kg, could increase TRX levels in nasal lavage fluids.

### 3.7. Influence of Quercetin on the Development of OVA-Induced Nasal Allergy-Like Symptoms

The final set of experiments was performed to examine whether the oral administration of quercetin in OVA-sensitized mice could inhibit the development of nasal allergy-like symptoms, which were induced by the nasal antigenic challenge. Nasal symptoms were assessed by counting the number of sneezes and nasal rubbing movements for 10 min immediately after the OVA nasal challenge. As shown in Figure 7, treating the OVA-sensitized mice with less than 15.0 mg/kg of quercetin could not inhibit the development of nasal allergy-like symptoms: the number of sneezes and nasal rubbing movements were nearly identical (not significant) to those observed in the non-treated controls. Conversely, the oral administration of more than 20.0 mg/kg of quercetin attenuated the development of nasal allergy-like symptoms, and the number of sneezes and nasal rubbing movements was significantly lower than those observed in the non-treated controls (Figure 7).

## 4. Discussion

The results obtained from the in vitro experiments clearly show that quercetin can increase the ability of HNEpCs to produce TRX in response to H_2_O_2_ stimulation. The minimum concentration that caused a significant increase in TRX production was 1.0 nM.

After the oral administration of 64 mg of quercetin to humans, quercetin plasma levels gradually increased and attained peak at 650 nM, with a half-life elimination of 17–24 h [25]. Although there is no standard recommended dosage of quercetin, a dose of 1200 to 1500 mg per day is commonly used [26] as a supplement. It is also observed that a 1200 mg dose could lead to a plasma concentration of up to 12 μM [25], which is higher than the concentration necessary to induce the increase in the ability of HNEpCs to produce TRX in vitro. Based on these reports, the findings of the present in vitro study may reflect the biological function of quercetin in vivo. At present, we cannot exclude the possibility that the stimulation of thioredoxin production at higher concentrations of hydrogen peroxide and quercetin may be cellular protective mechanism against the cytotoxicity induced by these agents. Further experiments are needed to test this possibility.

AR is defined as an allergic inflammation of the nasal mucosa and is characterized by a symptom complex that consists of any combination of sneezing, nasal congestion, and nasal itching, among others [1,2]. These symptoms are primarily induced by chemical mediators from mast cells, such as histamine, tryptase, and kinin [1,2]. These mediators also recruit other inflammatory cells, including neutrophils and eosinophils, to the mucosa [1]. These polymorphonuclear leukocytes secrete harmful granular proteins and ROS, which cause tissue remodeling and persistent AR [18,27]. Because ROS are necessary for life, the body initiates several mechanisms to decrease ROS-induced tissue damage and to repair damage that occurs, including several enzymes and proteins [19]. Among these mechanisms, TRX attracts attention as not only an important anti-oxidative factor but also as a protective factor in the development of various inflammatory diseases, including AR [22,23]. TRX is reported to suppress eosinophil chemotaxis induced by CC chemokine stimulation through the suppression of both the activation of extracellular signal-regulated kinase 1/2 and p38 mitogen-activated protein kinase pathways [28]. Treating mice with TRX inhibits the development of airway inflammation and the overproduction of macrophage inflammatory protein (MIP)-1, RANTES, IL-4, and IL-5, which are responsible for the development of allergic inflammatory responses [22,23]. Furthermore, airway remodeling and eosinophilic inflammation induced by chronic antigen exposure were prevented in TRX transgenic mice that displayed overproduction of TRX [23]. Together with these reports, the present results obtained in in vivo experiments suggest that quercetin increases TRX production in the nasal mucosa and results in a favorable modification of the clinical conditions of AR. However, before concluding that the oral administration of quercetin in AR patients increases the ability of nasal cells, particularly epithelial cells, to produce TRX and attenuate the development of AR, we must examine the influence of quercetin on TRX production in vivo. Therefore, the second half of the study was performed to examine whether quercetin could also increase the ability of nasal cells to produce TRX after specific allergen inhalation and whether this activity was related to the development of nasal allergy-like symptoms in OVA-sensitized mice. The present in vivo data showed that nasal lavage fluids obtained from sensitized-non-treated mice contained higher levels of lipid peroxide compared to those from non-sensitized mice. Moreover, the oral administration of quercetin decreased lipid peroxide levels and increased TRX levels in nasal lavage fluids. Furthermore, the oral administration of quercetin to OVA-sensitized mice inhibited the development of nasal allergy-like symptoms after the OVA nasal challenge. The minimum concentration that caused significant changes in these parameters was 20 mg/kg. From these results, it can be reasonably interpreted that the actions of quercetin on TRX production may represent a possible mechanism that can explain the favorable effects of quercetin on AR.

The present data clearly show that quercetin enhances the ability of nasal cells to produce TRX in response to stimulation with either H_2_O_2_ or specific allergens in vitro and in vivo, despite the suppression of TRX mRNA expression. Furthermore, our previous report clearly showed that quercetin inhibited the production of chemokines, such as eotaxin and macrophage inflammatory protein-1beta (MIP-1β), by suppressing the mRNA expression of chemokines in eosinophils after stem cell factor simulation [29]. Furthermore, quercetin exerts suppressive effects on the activation of transcription factors, which are essential for several types of endogenous immune-modulatory proteins [30]. Synthesis of proteins in cells requires two quite different steps: in transcription, the first step, specific mRNA is synthetized from DNA in the nucleus. The newly synthetized mRNA travels through the nuclear membrane into the cytoplasm where it binds to mRNA-binding sites on ribosomes and initiates protein synthesis, which is called translation. From these established concepts, there is a possibility that quercetin increases the translatable activity of TRX mRNA and results in the production and secretion of large amounts of TRX from nasal epithelial cells after stimulation. Although glucocorticoids, which are considered first-line therapeutic agents in the treatment of AR [2], are accepted to exert their immune-modulatory effects by suppressing inflammatory mediator mRNA expression, they can increase the ability of cells to produce an immune-modulatory peptide, uteroglobin, after inflammatory stimulations by enhancing the translation of uteroglobin mRNA [31,32]. These reports support the speculation that the translation of TRX mRNA is enhanced by quercetin and results in the appearance of a large amount of TRX in both culture supernatants and nasal secretions.

Oral allergy syndrome (OAS), also recognized as pollen-food syndrome, is an allergic response in the oral cavity following the ingestion of fruits, vegetables, or nuts. OAS reportedly occurs in approximately 20–70% of patients with AR and atopy [33]. Pollen-specific IgE antibodies in AR patients recognize homologous dietary allergens that share the same epitopes of pollen and trigger the cross-reaction between allergens in pollens and those in foods, resulting in the development of OAS [33]. OAS includes several allergic reactions that occur very rapidly, within minutes of eating a trigger food. The most common symptoms are itchy mouth, scratchy throat, or swelling of the lips, tongue, and throat [33,34]. Although no standard treatment for OAS exists, antihistamines and oral steroids can help relieve symptoms [33], which suggests that quercetin will be a good candidate to supplement the treatment of OAS.

## 5. Conclusions

The results obtained from the present experiments strongly suggest that quercetin increases the ability of nasal epithelial cells, to produce TRX after stimulation with oxidants or allergens. Moreover, quercetin results in the attenuation of development of the clinical symptoms of AR by suppressing oxidative stress responses in nasal mucosa.

## Figures and Tables

**Figure 1 medicines-05-00124-f001:**
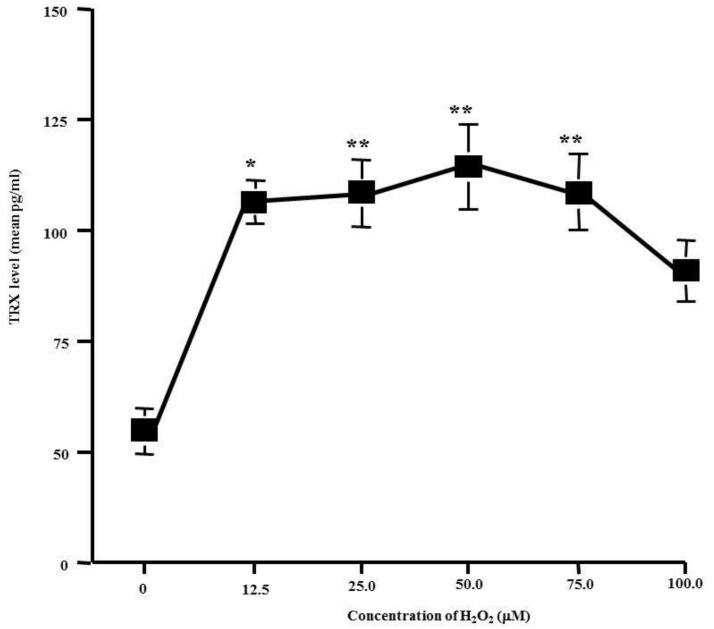
Influence of H_2_O_2_ on thioredoxin (TRX) production from HNEpCs in vitro. Nasal epithelial cells (5 × 10^5^ cells) were stimulated with various concentrations of H_2_O_2_. After 24 h, TRX levels in culture supernatants were examined with ELISA. The data are expressed as the mean pg/mL ± SE of triplicate cultures. One representative experiment of two is shown in this figure. *: *P* < 0.05 versus control (0); ** *P* > 0.05 versus 12.5 μM H_2_O_2_.

**Figure 2 medicines-05-00124-f002:**
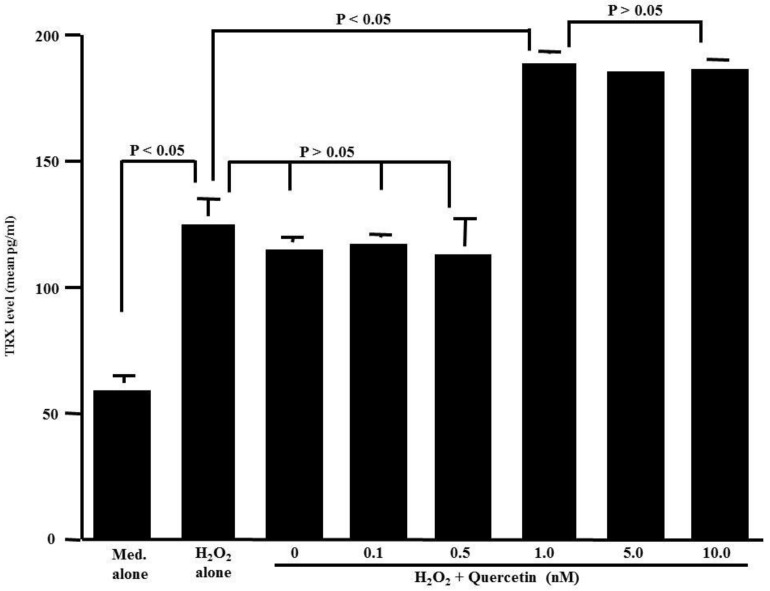
Influence of quercetin on thioredoxin (TRX) production from HNEpCs induced by H_2_O_2_ stimulation in vitro. Nasal epithelial cells (5 × 10^5^ cells) were stimulated with 50 μM H_2_O_2_ in the presence of various concentrations of quercetin for 24 h. TRX levels in culture supernatants were examined with ELISA. The data are expressed as the mean pg/mL ± SE of triplicate cultures. One representative experiment of two is shown in this figure. Med. alone: Medium alone.

**Figure 3 medicines-05-00124-f003:**
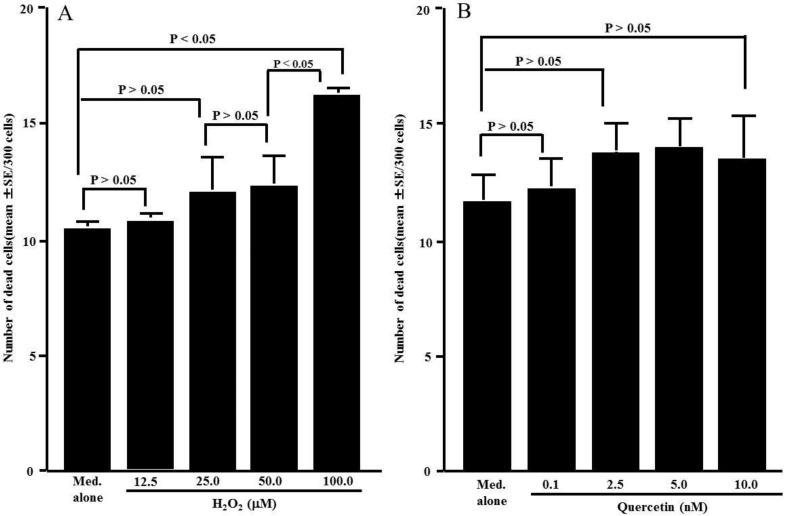
Influence of H_2_O_2_ (**A**) and quercetin (**B**) on cell viability. Nasal epithelial cells (5 × 10^5^ cells) were stimulated with various concentrations of either H_2_O_2_ or quercetin for 24 h. The trypan blue exclusion test was performed, and the number of dead cells was counted out of 300 total cells. The data are expressed as the mean number of dead cells ± SE of triplicate cultures. One representative experiment of two is shown in this figure. Med. alone: Medium alone.

**Figure 4 medicines-05-00124-f004:**
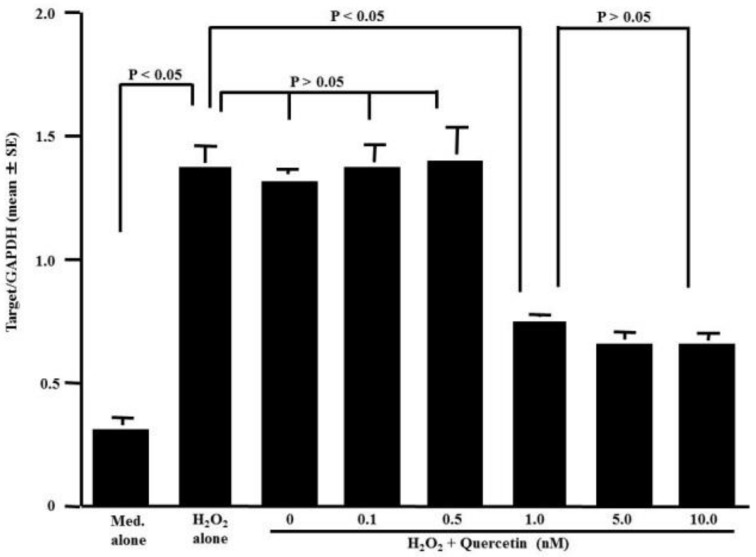
Influence of quercetin on TRX mRNA expression in HNEpCs. Nasal epithelial cells (5 × 10^5^ cells) were stimulated with 50 μM H_2_O_2_ in the presence or absence of quercetin for 12 h. TRX mRNA levels in the cultured cells were examined by RT-PCR. The data are expressed as the mean Target/GAPD ± SE of triplicate cultures. One representative experiment of two is shown in this figure. Med. alone: Medium alone.

**Figure 5 medicines-05-00124-f005:**
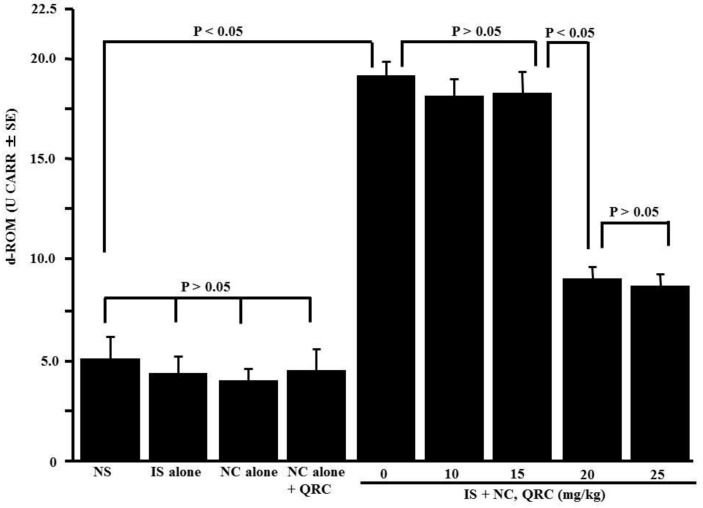
Influence of quercetin on the appearance of lipid peroxide in nasal lavage fluids after ovalbumin (OVA) sensitization in mice. BALB/c mice were sensitized by an intraperitoneal injection of OVA on days 0, 7, and 14. Seven days after the final sensitization, the OVA-sensitized mice were intranasally challenged with OVA on days 21, 23, and 25, and various concentrations of quercetin were administered orally once a day for five consecutive days. Nasal lavage fluids were obtained from mice 6 h after the OVA nasal challenge. Lipid peroxide levels were measured using the d-ROM test. The data are expressed as the mean CARR U ± SE of five mice. NS: non-sensitized; IS: intraperitoneal sensitization; NC: nasal challenge alone; QRC: quercetin.

**Figure 6 medicines-05-00124-f006:**
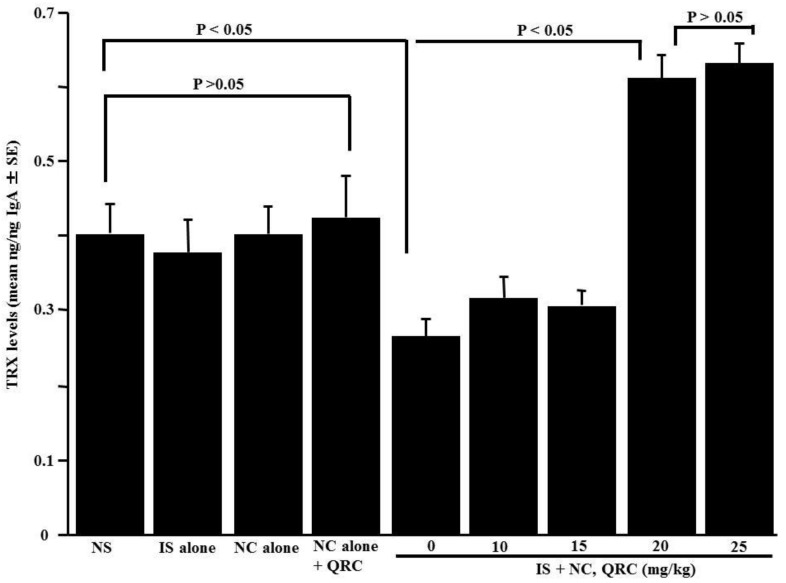
Influence of quercetin on the appearance of thioredoxin (TRX) in nasal lavage fluids obtained from OVA-sensitized mice after the OVA nasal challenge. BALB/c mice were sensitized by an intraperitoneal injection of OVA on days 0, 7, and 14. Seven days after the final sensitization, the OVA-sensitized mice were intranasally challenged with OVA on days 21, 23, and 25, and various concentrations of quercetin were administered orally once a day for five consecutive days. Nasal lavage fluids were obtained from the mice 6 h after the nasal antigenic challenge. TRX levels were examined by an ELISA. The data are expressed as the mean ng/ng IgA ± SE of five mice. NS: non-sensitized; IS: intraperitoneal sensitization; NC: nasal challenge alone; QRC: quercetin.

**Figure 7 medicines-05-00124-f007:**
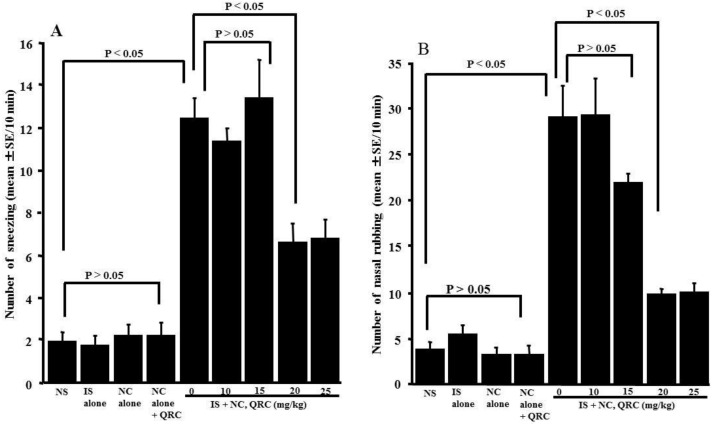
Influence of quercetin on the development of nasal allergy-like symptoms in OVA-sensitized mice after the OVA nasal challenge. BALB/c mice were sensitized by an intraperitoneal injection of OVA on days 0, 7, and 14. Seven days after the final sensitization, the OVA-sensitized mice were intranasally challenged with OVA on days 21, 23, and 25, and various concentrations of quercetin were administered orally once a day for five consecutive days. Nasal allergy-like symptoms, the number of sneezes (**A**), and nasal rubbing behaviors (**B**) were counted for 10 min immediately after the final nasal antigenic challenge. The data are expressed as the mean ± SE of five mice. NS: non-sensitized; IS: intraperitoneal sensitization; NC: nasal challenge alone; QRC: quercetin.

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
