# Peer review of "Quercetin Enhances the Thioredoxin Production of Nasal Epithelial Cells In Vitro and In Vivo"

_medicines, 2018, doi:10.3390/medicines5040124_

Round 1

Reviewer 1 Report

The present paper is interesting as it reports the bioactivity of quercetin on nasal epithelial cells in in vitro and in vivo systems. However, I have some suggestions for the authors to improve the paper:

a) in introduction the authors should better describe the antiradical properties of quercetin, being the most import biological effect reported in this work. I suggest to cite some recent works, including Biomimetics, 2017, 2(3), 9; Vitis: Journal of Grapevine Research, 2017, 56(1), 19-26; Journal of Functional Foods, 2018, 40, 68-75.

b) Authors report “The data are expressed as the mean pg/ml ± 191 SE of triplicate cultures. The experiments repeated twice with similar results”.. what does it mean? The experiments were performed in triplicate or duplicate? Please, explain better.

c) In in vitro systems, authors should perform both MTT and trypan blue exclusion test because they must demonstrate that the high doses of H2O2 (and quercetin) were not toxic for cells, as we can expect knowing the strong oxidative activity of this ROS and the potential toxicity of the plant metabolite.

d) the level of TRX mRNA should be also measured to confirm the observed data about protein level.

Author Response

I greatly appreciate your valuable comments. My replies to your specific comments are as follows. Revised portions are marked with underlines.

Comment a: According to this comment, I inserted recent two references (No. 9 and 10) into Introduction section (Page 2, Line 58).

Comment b: According to this comment, I revised figure legends ((Page 9, line 236; Page 10, line 244).

Comment c and d: According to these comments, I performed additional experiments, in which cytotoxicity of substances and mRNA expression were examined. Materials and Methods section was revised (Page 4, Lines 107-108; Page 4, Lines 111 to Page 5, Lines 129). Results section was also revised (Page 7, Lines 186 to Page 8, Lines 201).  New two figures (Figure 3 and 4) were created by using new data.

Reviewer 2 Report

medicines-381833. In the present study, the authors demonstrated that quercetin attenuates allergic rhinitis by enhancing thioredoxin (TRX) production. The results are interesting, and may provide readers and investigators with good information.

1) There are only analytical data of TRX in nasal lavage fluids, in addition to clinical symptoms. It is strongly recommended that morphological analysis (microphotographs) for the pathological changes should be included.

2) In addition, cytokines and inflammatory cells, especially eosinophils, in the nasal lavage fluids must be analyzed.

3) Please comment on the negative effects in low concentrations (<1 nM) of quercetin.

Author Response

I greatly appreciate your valuable comments. My replies to your specific comments are as follows. Revised portions are marked with underlines.

Major comments

Comment 1, 2 and 3: As stated in the manuscript, the present in vivo experiments require 25 day after starting sensitization. However, I received a notification from editorial office of Medicines that I must re-submit the revised version of the manuscript by 10 November, 2018. So, I cannot perform additional in vivo experiments according to your suggestions. My experimental group is now performing new experiments, in which molecular mechanisms of quercetin on thioredoxin production by using low concentration of the agent are investigated. After getting new data, I shall describe the manuscript and submit to your journal near future. Thank you for your valuable comments.

Other changes:

1) According to the comments raised by Reviewer 2, I inserted recent papers in Introduction section (Page 2, Line 56).

2) I revised figure legends (Page 9, line 236; Page 10, line 244).

3) According to the comments raised by Reviewer 2, I examined cytotoxicity of H2O2 and quercetin in vitro. To do this, Materials and Methods was revised (Page 4, Lines 111-114). I also examined the influence of quercetin on TRX mRNA expression in vitro. To do this, Materials and Methods section was revised (Page 4, Lines 107-108; Page 4, Lines 111 to Page 5, lines 129).

4) To perform additional experiments, I inserted a new reference (No. 24).

5) The results obtained from additional experiments were stated in Results section (Page 7, Lines 187 to Page 8, Lines 202) and new two figures (Figure 3 and 4) were created by using new data.